# Skeleton-bridged Point Completion: From Global Inference to Local Adjustment

**Yinyu Nie**[1,2,†]    **Yiqun Lin**[2]    **Xiaoguang Han**[2,*]    **Shihui Guo**[3]
**Jian Chang**[1]    **Shuguang Cui**[2]    **Jian Jun Zhang**[1]
[1]Bournemouth University    [2]SRIBD, CUHKSZ    [3]Xiamen University

## Abstract

Point completion refers to complete the missing geometries of objects from partial point clouds. Existing works usually estimate the missing shape by decoding a latent feature encoded from the input points. However, real-world objects are usually with diverse topologies and surface details, which a latent feature may fail to represent to recover a clean and complete surface. To this end, we propose a skeleton-bridged point completion network (**SK-PCN**) for shape completion. Given a partial scan, our method first predicts its 3D skeleton to obtain the global structure, and completes the surface by learning displacements from skeletal points. We decouple the shape completion into structure estimation and surface reconstruction, which eases the learning difficulty and benefits our method to obtain on-surface details. Besides, considering the missing features during encoding input points, SK-PCN adopts a local adjustment strategy that merges the input point cloud to our predictions for surface refinement. Comparing with previous methods, our skeleton-bridged manner better supports point normal estimation to obtain the full surface mesh beyond point clouds. The qualitative and quantitative experiments on both point cloud and mesh completion show that our approach outperforms the existing methods on various object categories.

## 1   Introduction

Shape completion shows its unique significance in fundamental applications such as 3D data scanning & acquisition and robot navigation. It focuses on completing the missing topologies and geometries of an object from partial and incomplete observations, e.g., point clouds captured under occlusion, weak illumination or limited viewpoints. Unlike image completion [15, 44, 23] that is well addressed with CNN-based approaches, point clouds present inherent irregularity and sparseness that challenge the direct application of grid-based convolutions to 3D shape completion.

Deep learning methods attempt to achieve this target with various representations, e.g., points [28, 42, 32, 14, 36, 20, 40, 37], voxels [10, 2, 35, 8, 12] or implicit fields [19, 29, 22, 5]. Voxels discretize the shape volume into 3D grids. It preserves shape topology but fine-detailed voxel quality relies on high resolution, improving which exponentially increases the time consumption. Implicit fields represent shapes with a signed distance function (SDF). Theoretically it can achieve arbitrary resolution though, learning an accurate SDF still relies on the quality of voxel grids, and these methods require massive spatial sampling to obtain an SDF for a single object, which distinctly increases the inference time [19, 29, 22, 5]. Besides, both voxel and SDF methods do not preserve the surface information and present defective results on complex structures. Point cloud is a natural representation of shapes that discretizes the 2-manifold surface. Comparing with voxels and SDFs, 3D points are more controllable, scalable and efficient for learning, which makes it popular for shape completion. However, existing methods commonly adopt an encoder&decoder to parse 3D points [42, 32], making

---

[†] Work done during internship at Shenzhen Research Institute of Big Data, The Chinese Univerity of Hong Kong, Shenzhen (SRIBD, CUHKSZ).
[*] Corresponding author: `hanxiaoguang@cuhk.edu.cn`

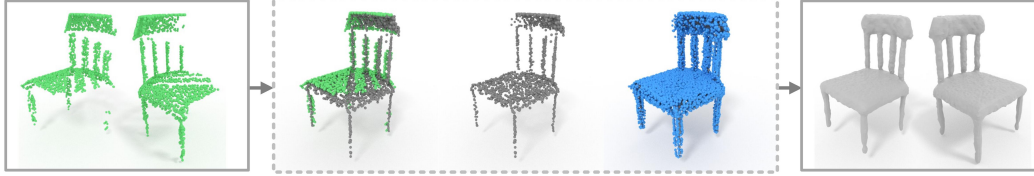

Figure 1: From a partial scan of an object (green points), SK-PCN estimates its meso-skeleton (grey points) to explicitly extract the global structure, and pairs the local-global features for displacement regression to recover the full surface points (blue points) with normals for mesh reconstruction (right).

them struggle to keep shape topology and produce coarse results. Mesh-based methods recover ordered surface points, but current methods predict object meshes by deforming templates (e.g., meshed spheres or planes [11]), making it restricted from recovering complex structures.

To preserve the shape structure and complete surface details, we provide a new completion manner, namely **SK-PCN**, that maps the partial scan to the complete surface bridged via the *meso-skeleton* [38] (see Figure 1). Recovering the missing structure and details from an incomplete scan generally requires both global and local features. Instead of using encoders to extract a latent layer response as the global feature [41, 14, 42, 36], we explicitly learn the meso-skeleton as the global abstraction of objects, which is represented by 3D points located around the medial axis of a shape. Comparing with surface points, meso-skeleton holds a more smooth and compact shape domain, making networks easier to be trained. It also keeps the shape structure that helps predict topology-consistent meshes.

To further recover surface details, prior works usually expand the global feature with upsampling [41, 18] or skip connections [36, 37] by revisiting local features from previous layers. Our method completes shape details with an interpretable manner, that is learning to grow surface points from the meso-skeleton. Specifically, SK-PCN dually extracts and pairs the local features from the partial scan and the meso-skeleton under multiple resolutions, to involve corresponding local features to skeletal points. As local structures are commonly with repetitive patterns (e.g., table legs are usually with the same geometry), we bridge these local-global feature pairs with a **Non-Local Attention** module to select and propagate those contributive local features from the global surface space onto skeletal points for missing shape completion. Moreover, to preserve the fidelity on observable regions, we devise a local refinement module and a patch discriminator to merge the original scan to the output. Unlike previous methods where point coordinates are directly regressed, we complete the surface by learning displacements from skeletal points. It is not only because learning residuals is easier for training [13]. These displacement values show high relevance with surface normals [38], which better supports the point normal estimation for our mesh reconstruction.

Our contributions are three-fold. First, we provide a novel learning modality for point completion by mapping partial scans to complete surfaces bridged via meso-skeletons. This intermediate representation preserves better shape structure to recover a full mesh beyond point clouds. Second, we correspondingly design a completion network SK-PCN. It end-to-end aggregates the multi-resolution shape details from the partial scan to the shape skeleton, and automatically selects the contributive features in the global surface space for shape completion. Third, we fully leverage the original scan for local refinement, where a surface adjustment module is introduced to fine-tune our results for a high-fidelity completion. Extensive experiments on various categories demonstrate that our method outperforms previous methods and reaches the-state-of-the-art.

## 2 Related Work

In this section, we mainly review the recent development of 3D deep learning on shape generation, shape completion and skeleton-guided surface generation.

**Shape Generation.** Shape generation aims at predicting a visually plausible geometry from object observations (e.g., images, points and depth maps). Some architectures support shape generation conditioned on various input sources by changing the encoder, where 3D shapes are decoded from a latent vector and represented by points [9], voxels [10, 7, 8], meshes [34, 11, 31, 24] or an SDF [22, 5, 19]. They share the similar modality, that is to decode the equal-size bottleneck feature for shape prediction. This implicit manner reveals the limitation of producing an approximating shape to the target. Different from them, we aim at shape completion that preserves the original shape information and completes the missing geometries with high fidelity.

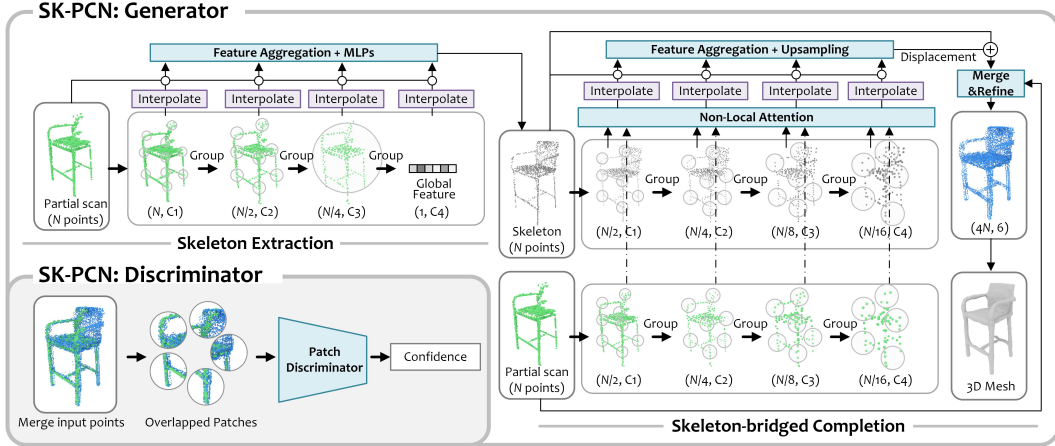

Figure 2: Network architecture of our method. SK-PCN consists of a shape generator and a patch discriminator. The shape generator produces a meso-skeleton first, and uses it to aggregate the multi-resolution local features on the global surface space for surface completion. The patch discriminator measures the fidelity score of our completion results on the overlapped area with the input scan. The layer specifications are detailed in the supplementary material.

**Shape Completion.** Shape completion aims to recover the missing shape from partial point cloud. Since voxel/SDF-based methods rely on high-quality voxel grids, and are with relative low inference efficiency [8, 29, 19, 22, 5], more works focus on point cloud completion [42, 32, 40, 14, 20, 36, 37]. Similar to shape generation, several point completion methods [40, 41] leverage a 3D encoder-decoder structure, especially after the pioneer work PointNet and PointNet++ [26, 27]. However, directly decoding the bottleneck feature from encoders shows inadequacy in expressing details. From this, [42, 36, 37] use skip or cascaded connections to revisit the low-level features to extend shape details. [20, 42] adopt a coarse-to-fine strategy to decode a coarse point cloud and refine it with dense sampling or deforming. PF-Net [14] designs a pyramid decoder to recover the missing geometries on multiple resolutions. However, implicitly decoding a latent feature does not take into account the topology consistency. The recent P2P-Net [40] learns the bidirectional deformation between the input scan and complete point cloud. It achieves compact completion results but still struggles to recover the topology especially on invisible areas. Our method implements the shape completion in an explicit manner. The structure of 3D shapes is preserved with skeletal points which guide the surface completion to predict globally and locally consistent shapes.

**Skeleton-guided Surface Generation.** Using shape skeletons to guide surface recovery has been well developed with traditional optimization strategy, wherein [30, 3, 38] associate the surface points with its skeleton to represent a compact and smooth surface. Deep learning methods receive rising attention with the advance of shape representation. However, previous methods more focus on learning the skeleton of a specific shape (e.g., for hand pose estimation [1] or human body reconstruction [16]). The recent work [31] provides a solution to infer 3D skeleton from images which also bridges and benefits the learning of single-view surface reconstruction. P2P-Net [40] supports bidirectionally mapping between skeletal points and surface points. Differently, we learn shape skeletons from partial scans as an intermediate representation to guide surface completion.

## 3 Method

We illustrate the architecture of SK-PCN in Figure 2. Given a partial scan, we aim at completing the missing geometries while preserving fidelity on the observable region. To this end, our SK-PCN is designed with a generator for surface completion, and a patch discriminator to distinguish and refine our results with the ground-truth. The generator has two phases: skeleton extraction and skeleton-bridged completion. The skeleton extraction module groups and parallelly aggregates the multi-resolution feature from the input to predict the skeletal points. The completion module shares the similar feature extraction process. It dually obtains multi-resolution features from both the skeleton and the input, and pairs them on each resolution scale (see Figure 2). For each pair, a Non-Local Attention module is designed to search the contributive local features from the partial scan to each skeletal point. These local features are then interpolated back to the skeletal points

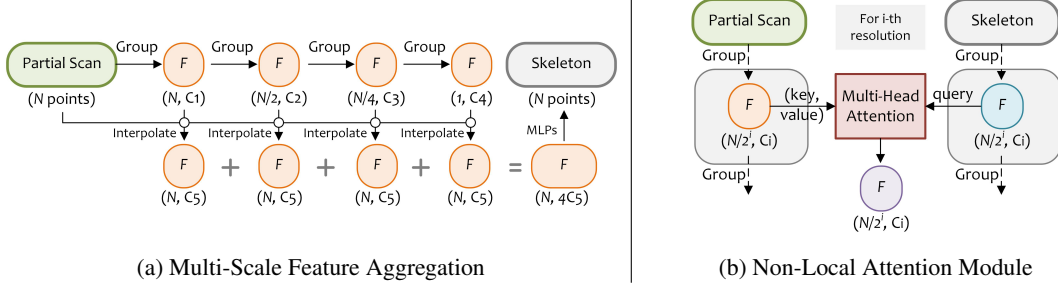

| (a) Multi-Scale Feature Aggregation | (b) Non-Local Attention Module |

Figure 3: Illustration of the multi-scale feature aggregation for our skeleton extraction (a) and the Non-Local Attention module to broadcast local details from the partial scan to skeletal points (b).

and aggregated to regress their displacements to the shape surface with the corresponding normal vectors on the surface. To preserve the shape information of the observable region, we merge the input to our shape followed with surface adjustment and produce the final mesh with Poisson Surface Reconstruction [17]. The details of each submodule are elaborated as follows.

## 3.1 Learning Meso-Skeleton with Global Inference

Meso-skeleton is an approximation of the shape medial axis. In our work, the ground-truth meso-skeletons are calculated with [38], and we represent them by 3D points for learning. As skeletons only keep shape structure, they do not preserve surface details. To this end, we devise a multi-scale feature aggregation to obtain point features under different resolutions (see Figure 3a). We adopt the set abstraction layers of [27] to progressively group and downsample point clouds to the coarser scale and obtain the global feature. Afterwards, these multi-scale point features are interpolated back to the partial scan with the feature propagation [27]. Then we concatenate them together to regress the skeletal point coordinates with MLPs. It attaches global features from different resolutions to the partial scan and relies on the network to select the useful ones for skeletal point regression.

## 3.2 Skeleton2Surface with Non-local Attention

Different from learning skeletons where the global feature takes the primary role, surface completion from a shape skeleton focuses on keeping the observable region and completing missing details.

**Non-Local Attention.** The insight of our method is that: shape skeletons show the spatial structure, which informs about the missing regions and guides the completion by leveraging the observable information. On this point, SK-PCN revisits the input scan to provide skeletal points with local details (see Figure 2) to recover the missing shape from different resolutions. Specifically, it dually extracts the multi-scale features from both the skeleton and input scan using the same feature aggregation module in Figure 3a. We do so to make skeletal points able to extract input details on different resolutions. Besides we observe that man-made objects are commonly with repetitive patterns (e.g., table legs are usually with the same structure). To this end, we design a Non-Local Attention module (see Figure 3b) to selectively and globally propagate local features to skeletal points. Specifically, on the $i$-th resolution, we denote the point features from the input and the skeleton by $\mathbf{P}_i, \mathbf{Q}_i \in \mathbf{R}^{\left(N/2^i, C_i\right)}$. $N$ is the input point number, and $C$ denotes its feature length. Here we adopt the attention strategy [33] to search the correlated local feature for each skeletal point in $\mathbf{Q}_i$ with:

$$\mathbf{Q}_i^* = \mathtt{softmax}\left(\frac{\mathtt{dot}(\mathbf{P}_i W_p, \mathbf{Q}_i W_q)}{\sqrt{d_i}}\right)\mathbf{P}_i, \tag{1}$$

where $W_p, W_q \in \mathbf{R}^{(C_i, d_i)}$ are the weights to be learned. $\mathtt{dot}\,(*,*)$ measures the feature similarity between the skeleton and input points. Thus for skeletal points in $\mathbf{Q}_i$, it selects and combines those useful point features from the partial scan $\mathbf{P}_i$ as the updated skeleton feature $\mathbf{Q}_i^*$. In practice, we adopt the multi-head attention strategy [33] to consider different attention responses. In our ablative study, we demonstrate that this module brings significant benefits in searching local features.

**Learning Surface from Skeleton.** After obtaining the multi-resolution local features for each skeletal point, we interpolate them back to the original skeleton and concatenate together (same to Figure 3a). Thus each skeletal point is loaded with multi-level local features. After that, we upsample the $N$ point features to four times denser with [41] to recover more surface details. Specifically, the point features $(N, C)$ are repeated with four copies, followed with grouped convolution [43] to

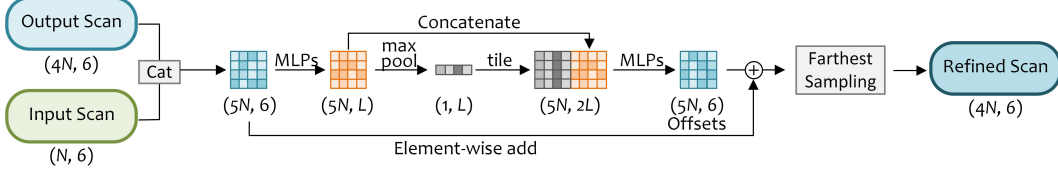

Figure 4: The pipeline of our surface adjustment module.

deform them individually and output a new feature matrix $(N, 4C)$. By reshaping it to $(4N, C)$, the upsampled points can be obtained with fully-connected layers. Rather than directly regressing point coordinates, we predict the displacements from skeletal points to the surface. It is because these displacement values show high relevance with surface normals [38], which better supports point normal estimation for our mesh reconstruction. Besides, learning residuals is beneficial to capture subtle details and also improves training efficiency.

## 3.3 Surface Adjustment with Local Guidance

As mentioned above, a completion method should preserve the geometry on the observable region. In this part, we design a generative adversarial module with a generator to merge input data to the output, and a discriminator to score the merging performance. The merging process is illustrated in Figure 4. We pre-calculate the normals of the input scan from coordinates $(N, 6)$ and concatenate them to our prediction $(4N, 6)$. The merged point cloud $(5N, 6)$ is followed with fully-connected layers and max-pooling to produce a shared feature. We tile and append this feature on each point to estimate offsets (inc. coordinates and normals) to the original output. However, merging input points results in denser distribution and defective boundaries on the overlapped area. For the first, we append the network with a farthest sampling layer to produce a uniformly-distributed point cloud with $4N$ points. For the second, to address the artifacts on boundaries, we adopt a patch discriminator to distinguish the merging result on the overlapped and boundary areas (see Figure 2). It randomly picks $m$ seeds on the input scan. For each seed, it groups $n$ points into a patch, which are located on the output scan within a radius $r$ to these seeds ($m = 24, n = 128, r = 1/10 \times$object size). For each patch, we utilize the basic architecture of [18] with a sigmoid layer to score the confidence value. It approximates 1 if the discriminator decides that a patch is similar to the ground-truth, and 0 if otherwise. Comparing with sampling patches over the whole surface, we observe that this method achieves better results in merging the input scan.

## 4 Loss Functions

In this section, we firstly define the point loss $\mathcal{L}_{\mathbf{P}}$ to compare the similarity between two point sets. Then a completion loss $\mathcal{L}_{\mathbf{C}}$ is provided to fulfill the surface completion. We denote the predicted/ground-truth skeletal points and surface points by $\mathbf{P}_k$ / $\mathbf{P}_k^*$ and $\mathbf{P}_s$ / $\mathbf{P}_s^*$ correspondingly.

**Point Loss.** Since the outputs consist of unordered points, Chamfer Distance $\mathcal{L}_{\mathrm{CD}}$ [9] is adopted to measure the permutation-invariant distance between two point sets. For normal estimation (only in surface completion), we use the cosine distance $\mathcal{L}_n$ [25] to compare two normal vectors. Besides, we also adopt a repulsion loss $\mathcal{L}_r$ to obtain evenly distributed points (similar to [41]). Thus for two point sets, we define the point loss $\mathcal{L}_{\mathbf{P}}$ between $\mathbf{P}$ and its ground-truth $\mathbf{P}^*$ by $\mathcal{L}_{\mathrm{CD}} + \lambda_n \mathcal{L}_n + \lambda_r \mathcal{L}_r$, where

$$\mathcal{L}_{\mathrm{CD}} = \sum_{x \in \mathbf{P}} \min_{y \in \mathbf{P}^*} \|x - y\|_2 / |\mathbf{P}| + \sum_{y \in \mathbf{P}^*} \min_{x \in \mathbf{P}} \|y - x\|_2 / |\mathbf{P}^*|, \tag{2}$$

$$\mathcal{L}_n = \sum_{x \in \mathbf{P}} (1 - \boldsymbol{n}_x \cdot \boldsymbol{n}_y) / |\mathbf{P}|, \ \ y \in \mathbf{P}_*, \tag{3}$$

$$\mathcal{L}_r = \sum_{x \in \mathbf{P}} \sum_{x_p \in N(x)} (d - \|x_p - x\|_2) / |\mathbf{P}|. \tag{4}$$

$\mathcal{L}_{\mathrm{CD}}$ in (2) presents the average nearest distance between $\mathbf{P}$ and $\mathbf{P}^*$. $|\mathbf{P}|$ denotes the point number in $\mathbf{P}$. In (3), $\boldsymbol{n}_y$ is the unit normal vector of the point in $\mathbf{P}^*$ that is the nearest neighbor to $x$. In (4), $\mathcal{L}_r$ requires the output points to be distant from each other and thus enforces a uniform distribution, where $N(x)$ are the neighbors of point $x$. $d$ is the maximal distance threshold ($d = 3e^{-4}$).

**Completion Loss.** Since SK-PCN predicts both skeleton and surface points, for each task, we adopt our point loss to measure their distance to the ground-truth. SK-PCN has three phases during shape

completion: 1. skeleton estimation; 2. skeleton2surface; and 3. surface adjustment. Thus we define the completion loss in surface generation by

$$\mathcal{L}_C \;=\; \lambda_k \mathcal{L}_{\mathbf{P}_k} + \lambda_s \mathcal{L}_{\mathbf{P}_s} + \lambda_m \mathcal{L}_{\mathbf{P}_s^m}. \tag{5}$$

In $\mathcal{L}_{\mathbf{C}}$, the $(\mathcal{L}_{\mathbf{P}_k}, \mathcal{L}_{\mathbf{P}_s}, \mathcal{L}_{\mathbf{P}_s^m})$ respectively correspond to the point losses from the three phases, where $\mathbf{P}_s^m$ is the refined version of $\mathbf{P}_s$ (see section 3.3). $\{\lambda_*\}$ are the weights to balance their importance.

**Adversarial Loss.** For the surface adjustment in section 3.3, we train our SK-PCN together with the patch discriminator using the least square loss [21, 18] as the adversarial loss:

$$\mathcal{L}_{\mathcal{G}} \;=\; \left[ D\left(\mathbf{P}_{patch}\right) - 1 \right]^2 \tag{6}$$

$$\mathcal{L}_{\mathcal{D}} \;=\; D\left(\mathbf{P}_{patch}\right)^2 + \left[ D\left(\mathbf{P}_{patch}^*\right) - 1 \right]^2, \tag{7}$$

where $D$ is the patch discriminator, $\mathbf{P}_{patch}$ and $\mathbf{P}_{patch}^*$ denote the estimated and ground-truth patch points on the overlapped area. A low $\mathcal{L}_{\mathcal{G}}$ means the discriminator scores our output with high confidence. We minimize the $\mathcal{L}_{\mathcal{D}}$ to make it able to distinguish our result with the ground-truth.

Overall, we train our SK-PCN end-to-end using the generator loss of $\mathcal{L}_C + \lambda_G \mathcal{L}_{\mathcal{G}}$ to implement shape completion, and the discriminator loss $\mathcal{L}_{\mathcal{D}}$ to preserve the fidelity on the observable region.

## 5 Results and Evaluations

### 5.1 Experiment Setups

**Datasets.** Two datasets are used for our training. 1) ShapeNet-Skeleton [31] for skeleton extraction, and 2) ShapeNetCore [4] for surface completion. We adopt the train/validation/test split from [39] with five categories (i.e., chair, airplane, rifle, lamp, table) and 15,338 models in total. For each object model, we align and scale them within a unit cube and obtain 8 partial scans by back-projecting the rendered depth maps from different viewpoints (see the supplementary file for data and split preparation). The full scan and its corresponding shape skeleton are used as supervisions.

**Metrics.** In our evaluation, we adopt the Chamfer Distance-$L_2$ (CD) [14] and Earth-Mover's Distance (EMD) [42] to evaluate completion results on surface points, and use CD together with normal consistency defined in [22] to test the quality of the estimated point coordinates and normals for our mesh reconstruction (see evaluations and comparisons on more metrics in the supplementary file).

**Implementation.** From skeleton estimation, skeleton2surface to surface adjustment, we first train each subnet of SK-PCN separately with fixing the former modules using our point loss. Then we train the whole network end-to-end with the generator and discriminator loss. We adopt the batch size at 16 and the learning rate at 1e-3, which decreases by the scale of 0.5 if there is no loss drop within five epochs. 200 epochs are used in total. The weights used in the loss functions are: $\lambda_k, \lambda_s = 1, \lambda_m = 0.1, \lambda_n = 0.001, \lambda_r = 0.1, \lambda_G = 0.01$. We present the full list of module and layer parameters, and inference efficiency in the supplementary material.

**Benchmark.** To investigate the performance of our method, we comprehensively compare our SK-PCN with state-of-the-art methods including MSN [20], PF-Net[14], PCN [42], P2P-Net [40], DMC [19], ONet [22] and IF-Net [5] on point cloud/mesh completion. We train all the models on the same dataset for a fair comparison. Inline with PF-Net[14], we benchmark the input scale with 2048 points, and the ground-truth with 10k points in evaluation.

Table 1: Quantitative comparisons on point cloud completion.

| Category | Chamfer Distance-$L_2$ ($\times 1000$) ↓ / Earth Mover's Distance ($\times 100$) ↓ | | | | | | |
| --- | --- | --- | --- | --- | --- | --- | --- |
| | DMC | MSN | PF-Net | P2P-Net | ONet | PCN | Ours |
| Airplane | 0.392 / 0.411 | 0.111 / 0.194 | 0.280 / 0.685 | 0.127 / 1.323 | 0.355 / 0.300 | 0.287 / 3.960 | **0.104 / 0.197** |
| Rifle | 0.337 / 0.631 | 0.086 / 0.107 | 0.213 / 0.913 | 0.045 / 0.850 | 0.281 / 0.294 | 0.190 / 3.927 | **0.033 / 0.082** |
| Chair | 0.383 / 1.057 | 0.322 / 0.541 | 0.581 / 2.090 | 0.294 / 3.125 | 1.426 / 1.552 | 0.530 / 3.228 | **0.255 / 0.486** |
| Lamp | 0.521 / 1.633 | 0.630 / 1.473 | 1.283 / 2.273 | 0.302 / 3.271 | 1.480 / 1.937 | 2.278 / 4.542 | **0.141 / 1.135** |
| Table | 0.442 / 1.083 | 0.498 / 0.639 | 0.933 / 3.165 | 0.374 / 3.005 | 1.439 / 1.230 | 0.700 / 3.098 | **0.343 / 0.594** |
| Average | 0.415 / 0.963 | 0.329 / 0.591 | 0.658 / 1.825 | 0.228 / 2.315 | 0.996 / 1.063 | 0.797 / 3.751 | **0.175 / 0.499** |

### 5.2 Comparisons with Point Completion Methods

We compare our method on point cloud completion with the baseline approaches including DMC [19], MSN [20], PF-Net[14], P2PNet [40], ONet [22] and PCN [42]. For all methods, the number of

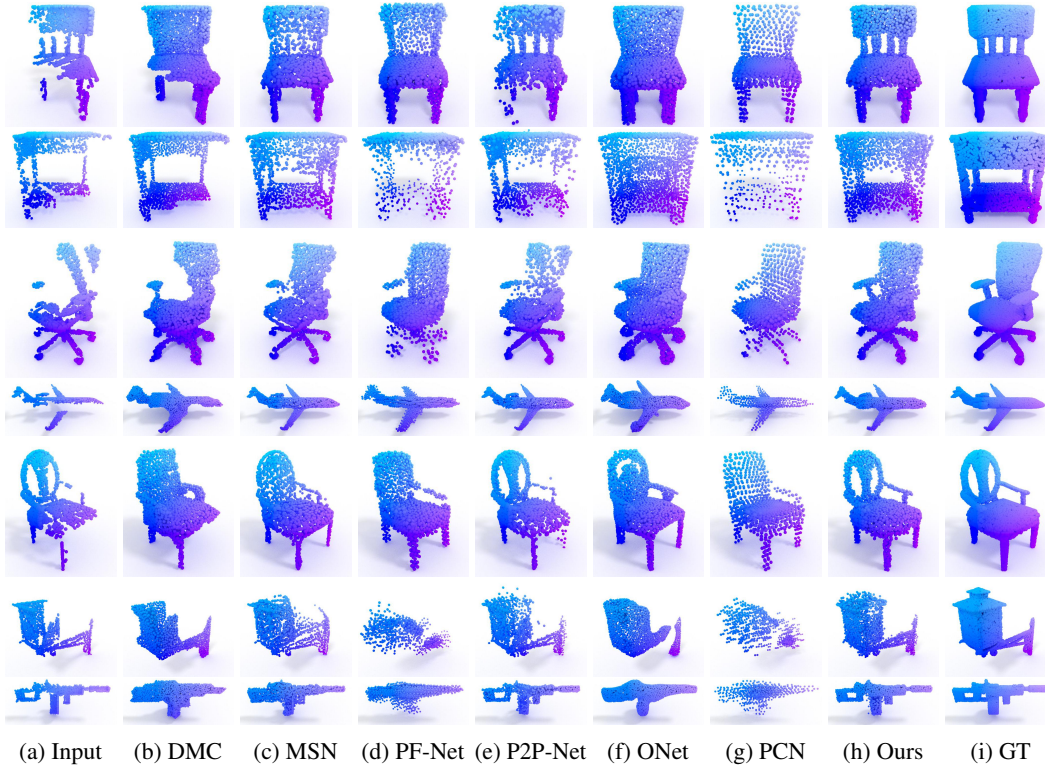

| (a) Input | (b) DMC | (c) MSN | (d) PF-Net | (e) P2P-Net | (f) ONet | (g) PCN | (h) Ours | (i) GT |

Figure 5: Comparisons on point cloud completion. From left to right respectively are: a) input partial scan; b) DMC [19]; c) MSN [20]; d) PF-Net [14]; e) P2P-Net [40]; f) ONet [22]; g) PCN [42]; h) ours; i) ground-truth scan.

Table 2: Quantitative comparisons on mesh reconstruction.

| Category | Chamfer Distance-$L_2$ ($\times 1000$) $\downarrow$ | | | | | Normal Consistency $\uparrow$ | | | | |
| | DMC | ONet | IF-Net | P2P-Net* | Ours | DMC | ONet | IF-Net | P2P-Net* | Ours |
|---|---|---|---|---|---|---|---|---|---|---|
| Airplane | 0.361 | 0.337 | 0.447 | 0.102 | **0.072** | 0.810 | 0.835 | 0.813 | 0.828 | **0.851** |
| Rifle | 0.326 | 0.272 | 0.297 | 0.035 | **0.022** | 0.682 | 0.747 | 0.857 | 0.831 | **0.925** |
| Chair | 0.328 | 1.400 | 0.745 | 0.258 | **0.159** | 0.781 | 0.770 | 0.824 | 0.801 | **0.863** |
| Lamp | 0.472 | 1.451 | 0.875 | 0.392 | **0.261** | 0.793 | 0.818 | 0.830 | 0.791 | **0.842** |
| Table | 0.280 | 1.405 | 0.910 | 0.321 | **0.246** | 0.838 | 0.826 | 0.846 | 0.810 | **0.881** |
| Average | 0.353 | 0.973 | 0.655 | 0.222 | **0.152** | 0.781 | 0.799 | 0.834 | 0.812 | **0.872** |

output points is set to 2,048 for a fair comparison (i.e., the upsampling rate is set to 1). We present the qualitative and quantitative results on the test set in Figure 5 and Table 1 respectively. From the results, we observe that the traditional encoder-decoder methods show inadequacy in preserving small structures (row 1 & 2) and original topology (row 3 & 5) when completing missing shapes. Using skeletons as guidance, our method better preserves the topology of shapes, where thin structures (row 1) and holes (row 5) are well recovered. Moreover, by merging the input information, our results achieve higher fidelity on the observable region (row 6 & 7). The quantitative results in Table 1 further demonstrate that we obtain superior scores in both coordinates approximation (CD values) and distribution similarity (EMD values).

## 5.3 Comparisons with Mesh Reconstruction Methods

As SK-PCN estimates point normals along with coordinates, we further evaluate our reconstructed meshes using Poisson Surface Reconstruction by comparing with the existing mesh completion methods including IF-Net [5], ONet [22] and DMC [19]. In this part, 8,192 points are uniformly sampled from each output to calculate the CD and normal consistency with the ground-truth (10k points with normals). Furthermore, we augment the P2P-Net [40] with normal prediction to investigate the completion performance without skeleton guidance (named by P2P-Net*). Specifically, we use the deformation module in P2P-Net to estimate the displacements from the input scan to the shape surface with point normals using extra channels (same to ours), and append the output layer with our upsampling module to keep a consistent number of points. We present the comparisons in Table 2 and

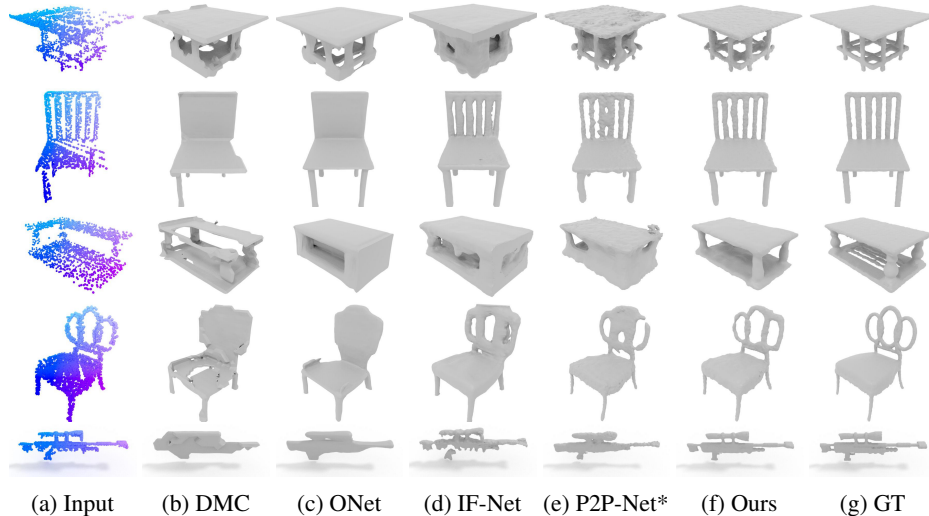

| (a) Input | (b) DMC | (c) ONet | (d) IF-Net | (e) P2P-Net* | (f) Ours | (g) GT |

Figure 6: Comparisons on mesh completion. From left to right respectively are: a) input partial scan; b) DMC [19]; c) ONet [22]; d) IF-Net [5]; e) P2P-Net* [40]; f) ours; g) ground-truth mesh.

Figure 6 (see more samples in the supplementary file). The results demonstrate that shape completion by decoding a latent feature (as in DMC [19], ONet [22] and IF-Net [5]) can produce an approximate and smooth shape but fail to represent small-scale structures. Besides, from Figure 6e, 6f and Table 2, we observe that using skeletal points as an intermediate representation significantly improves the normal estimation and produces local consistent normals (e.g., row 1, 2 & 4 in Figure 6f).

### 5.4 Ablative Analysis

To understand the effect of each module, we ablate our method with three configurations: $C_1$: w.o. Non-Local Attention & w.o. Surface Adjustment (Baseline); $C_2$: Baseline + Surface Adjustment; $C_3$: Baseline + Non-Local Attention; **Full:** Baseline + Non-Local Attention + Surface Adjustment. Note that the baseline method predicts the full scan by deforming via skeletal points without extra modules. It completes sur-

Table 3: Comparisons between ablated versions.

| Metric | $C_1$ | $C_2$ | $C_3$ | Full |
|---|---|---|---|---|
| CD $\downarrow$ | 0.340 | 0.293 | 0.205 | **0.175** |
| $CD_{comp}$ $\downarrow$ | 0.353 | 0.338 | 0.197 | **0.184** |
| $CD_{acc}$ $\downarrow$ | 0.326 | 0.248 | 0.212 | **0.166** |
| EMD $\downarrow$ | 2.261 | 1.013 | 0.725 | **0.499** |
| Normal Con. $\uparrow$ | 0.796 | 0.828 | 0.842 | **0.853** |

face points only using predicted skeletons. We devise this baseline to instigate how much the other modules leverage the input to improve the results. Here we output 2048 points for evaluation, and use the CD, normal consistency, EMD, completeness metric ($CD_{comp}$) and accuracy metric ($CD_{acc}$) to investigate the effects of each module. $CD_{comp}$ is defined with the average distance from each ground-truth point to its nearest predicted point, and $CD_{acc}$ is defined in the opposite direction. Their mean value is the Chamfer distance. We list the evaluations in Table 3 and visual results in Figure 7. The results indicate that the Non-Local Attention module manifests the most significant improvement of the overall performance ($C_3$ v.s. $C_1$). Merging input scan brings more gains in improving the $CD_{acc}$ values ($C_2$ v.s. $C_1$). It implies that merging a partial scan helps to extract more local information from the observable region, and combing the two modules achieves the best performance for shape completion.

We also investigate the significance of using 'skeleton' as the bridge for shape completion by replacing the skeletal points with 2,048 coarse surface points (see section 3.1). We implement this ablation on 'chair' category which presents sophisticated topology (see Figure 8). The CD$\downarrow$ and Normal Consistency $\uparrow$ values are 2.96e-4 and 0.81 respectively compared to our 1.59e-4 and 0.86. We think the reason could be that, coarse point cloud is still a type of surface points. Differently, skeletal points keep compact topology of the shape without surface details. Using it as a bridge makes our method easier to recover complex structures.

### 5.5 Discussions

In this section, we mainly discuss the performance on skeleton extraction with its impacts on the final results, and demonstrate the qualitative tests on real scans.

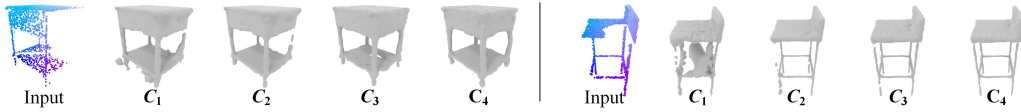

Figure 7: Mesh reconstruction with the configuration $C_1$, $C_2$, $C_3$ and **Full**.

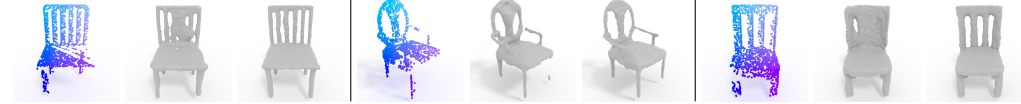

Figure 8: Skeleton v.s. Coarse points in shape completion. From left to right for each sample: input scan, results bridged with coarse points, and ours (bridged with skeletal points).

**Skeleton Extraction.** Since skeleton extraction performs significant role in our pipeline, we illustrate some quantitative samples of skeleton extraction in Figure 9 and the average CD value on all shape categories is 2.98e-4. Besides, we also find that the skeleton quality as a structure abstraction has an intuitive impact on the final results. For example, for a skeleton failed to represent some local structure (e.g. with unclear skeletal points), our Skeleton2Surface module will struggle to grow the counterpart surface, and we conclude these scenarios as our limitations (see Figure 10).

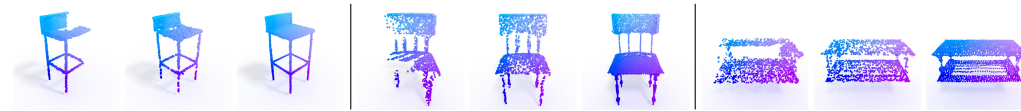

Figure 9: Skeleton extraction results. From left to right for each sample: input scan; predicted shape skeleton (2,048 points); and the ground-truth.

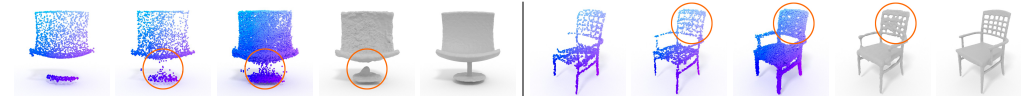

Figure 10: Limitation cases. From left to right for each sample: input partial scan, predicted skeleton, points and mesh, ground-truth mesh.

**Tests on Real Scans.** We also test our network (trained with ShapeNet) on real scans to investigate its robustness to real-world noises (see Figure 11). The input partial point clouds are back-projected from a single depth map and aligned to a canonical system [6]. From the results, we can observe that our method can achieve plausible results under different levels of incompleteness and noise.

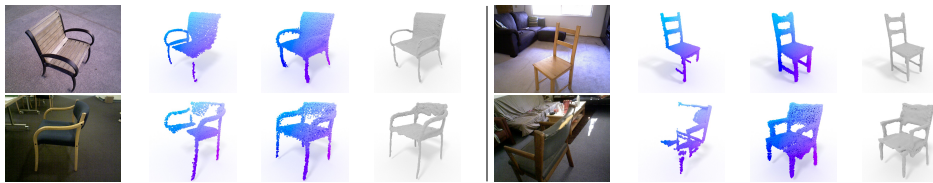

Figure 11: Tests on real scans [6]. From left to right: image of the target object; input partial scan; predicted point cloud; predicted 3D mesh.

## 6  Conclusion

In this paper, we present a novel learning modality for point cloud completion, namely SK-PCN. It end-to-end completes missing geometries from a partial point cloud by bridging the input to the complete surface via the shape skeleton. Our method decouples the shape completion into skeleton learning and surface recovery, where full surface points with normal vectors are predicted by growing from skeletal points. We introduce a Non-Local Attention module into point completion. It propagates multi-resolution shape details from the input scan to skeletal points, and automatically selects the contributive local features on the global shape surface for shape completion. Moreover, we provide a surface adjustment module to fully leverage input information and obtain high completion fidelity. Extensive experiments on both point cloud and mesh completion tasks demonstrate that our skeleton-bridged method presents high fidelity in preserving the shape topology and local details, and significantly outperforms the existing encoder-decoder-based methods.

## Broader Impact

Shape completion techniques have wide applications in industries, such as 3D data acquisition, shape surface recovery and robot navigation. Our research focuses on point cloud completion from scanning data. It shows benefits in improving the efficiency of 3D scanning in real-world environments, where objects are usually partly observable (e.g., occluded or with poor illumination conditions). Besides, it can also enhance the 3D scene reconstruction results from scanned data. On this base, It further helps collision detection for robots in automatic navigation. With the development of 3D cultural hetritage digitalization, it also shows potential capability in restoring 3D shapes of ancient artifacts. It also can be used as a 3D geometry restoration and repairing tool in computer graphics. We think there are no ethical or societal risks with this technique.

## Acknowledgments and Disclosure of Funding

The work was supported in part by the Key Area R&D Program of Guangdong Province with grant No. 2018B030338001, by the National Key R&D Program of China with grant No. 2018YFB1800800, by Shenzhen Outstanding Talents Training Fund, and by Guangdong Research Project No. 2017ZT07X152. It was also supported by NSFC-61902334, NSFC-61702433, NSFC-62072383, the Fundamental Research Funds for the Central Universities (20720190006), VISTA AR project (funded by the Interreg France (Channel) England, ERDF), Innovate UK Smart Grants (39012), the China Scholarship Council and Bournemouth University.

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
