[Supplementary Material]

# Skeleton-bridged Point Completion: From Global Inference to Local Adjustment
## *Supplementary Material*

**Yinyu Nie**[1,2,†]    **Yiqun Lin**[2]    **Xiaoguang Han**[2,*]    **Shihui Guo**[3]
**Jian Chang**[1]    **Shuguang Cui**[2]    **Jian Jun Zhang**[1]
[1]Bournemouth University    [2]SRIBD, CUHKSZ    [3]Xiamen University

# 1   Network Architecture and Parameter Setting

We provide the details of our network architecture and layer specifications in this section. In our paper, we adopt the same notations as [10]. The set abstraction layer is denoted by $SA\left(K, r, [l_1, l_2, ..., l_d]\right)$, and the feature propagation layer is represented by $FP\left([l_1, l_2, ..., l_d]\right)$. $K$ is the number of patches that are grouped from the input points. $r$ is the radius of the bounding ball for each patch (see Figure 2 in our paper). $[l_1, l_2, ..., l_d]$ represent the fully-connected layers inside the set abstraction and the feature propagation, where $l_i$ denotes the number of neurons in the $i$-th layer. Similarly, the fully-connected layers are represented by $MLP\left([l_1, l_2, ..., l_d]\right)$.

## 1.1   Learning Meso-Skeleton with Global Inference

We input our network with $N$ points and normals calculated from point coordinates, i.e., $(x, y, z, n_x, n_y, n_z)$. The parameter setting of our skeleton estimation network (see Section 3.1 of our paper) is illustrated in Figure 1, where $N$ denotes the number of input points. $R$ represents the scale of the 3D shape ($2R$ equals to the side length of the bounding box of the shape). $N \times (3 + d)$ means two outputs: the 3-dimensional point coordinates with corresponding $d$-dimensional point features. $N = 2048$ and $R = 0.5$.

Figure 1: Skeleton Estimation Network.

## 1.2   Skeleton2Surface with Non-local Attention

With the estimated shape skeleton, we propagate the surface features from the input scan to each skeletal point with our Non-Local Attention module (see Section 3.2 of our paper). Then the skeletal point features are aggregated to regress the displacements to the shape surface and the corresponding normal vectors on the surface. The network architecture is illustrated in Figure 2, wherein the

---

[†] Work done during internship at Shenzhen Research Institute of Big Data, The Chinese Univerisity of Hong Kong, Shenzhen (SRIBD, CUHKSZ).
[*] Corresponding author: `hanxiaoguang@cuhk.edu.cn`

Figure 2: Network Architecture of Skeleton2Surface.

upsampling layer is explained in Figure 3. The four parallel fully-connected layers in Figure 3 can be implemented with the efficient group convolutions [16].

Figure 3: Upsampling layer in Skeleton2Surface.

## 1.3 Surface Adjustment with Local Guidance

With the above layers, we can preliminarily obtain the surface points with normals. In Section 3.3 of our paper, we involve a surface adjustment to merge the input scan to improve the fidelity on observable regions. We present the surface adjustment network in Figure 4. For the discriminator, we utilize the basic architecture of [5] with a sigmoid layer to score the confidence value of each patch on our predicted surface. It approximates 1 if the discriminator decides that a patch is similar to the ground-truth, and 0 if otherwise.

Figure 4: Surface adjustment.

## 1.4 Running Time

We train our network with two TITAN-Xp GPUs and test it on a single GPU. The average time cost on point completion is 1.628 seconds per instance. The mesh reconstruction relies on an external Poisson Surface Reconstruction Library [9], which takes 1.412 seconds per instance on average.

## 2 Data Preparation

**Full scan data.** In our paper, we adopt the ShapeNet [1] dataset in our experiments. We observe that the man-made objects in ShapeNet are usually with non-manifold meshes and inner structures. To obtain watertight surface points and meshes of an object, we set up eight virtual cameras around the ShapeNet model to capture depth maps and reconstruct the surface mesh (see Figure 5). Specifically, we align and scale each model into a unit cube, and render the depth maps from eight viewpoints (centered at the eight corners of a cube with the side length at 2). We back-project the depth maps to 3D and obtain the ground-truth surface points. For points from each depth map, we also calculate their normal vectors with [12]. The direction of normal vectors are flipped outside the shape surface. The ground-truth surface mesh are reconstructed with Poisson Surface Reconstruction (PSR) [4]. The surface points and normals in the full scan data are used to supervise our surface point completion.

**Skeleton data.** We utilize the shape skeletons from ShapeNet-Skeleton dataset [11] to supervise our skeleton estimation network, where skeletal points are correspondingly aligned and scaled to the same scope with ShapeNet models.

Figure 5: Ground-truth data preparation

**Partial scan data.** In our training, we randomly select one partial scan from the eight viewpoints as the input data (see Figure 6), and use the full scan in Figure 5 as the ground-truth.

Figure 6: Input data preparation

We adopt the train/validation/test split from [13], which contains limited ShapeNet categories (containing airplane, chair, table, lamp, car in our experiments). For extra categories (inc. rifle, bench and watercraft), we adopt the split ratio of 6/2/2 in experiments.

## 3 More Qualitative Comparisons

We list more qualitative comparisons with previous methods, i.e., MSN [7], PF-Net[3], PCN [15], P2P-Net [14], DMC [6], ONet [8], IF-Net [2] and P2P-Net* (augment P2P-Net with normal estimation channels, see Section 5.3 in our paper) in Figure 7, where both the point and mesh completion results are compared on seven categories (inc. chair, lamp, rifle, table, airplane, bench and watercraft).

## 4 More Quantitative Comparisons

### 4.1 Comparisons on Extra Categories and Metrics

We compare our method on extra two categories (bench and watercraft) on both point and mesh completion in Table 1 and Table 2. In point completion evaluation, the number of output points is set

Figure 7: More qualitative comparisons on the testing set.

to 2,048 for a fair comparison. In Table 2, 8,192 points are uniformly sampled to evaluate the mesh reconstruction performance. We benchmark the input scale with 2,048 points, and the ground-truth with 10k points in both point and mesh evaluations.

Table 1: Quantitative comparisons on point cloud completion.

| Category | Chamfer Distance-$L_2$ ($\times 1000$) $\downarrow$ | | | | | Earth Mover's Distance ($\times 100$) $\downarrow$ | | | | |
|---|---|---|---|---|---|---|---|---|---|---|
| | MSN | PF-Net | P2P-Net | PCN | Ours | MSN | PF-Net | P2P-Net | PCN | Ours |
| Bench | 0.267 | 0.538 | 0.237 | 0.489 | **0.204** | **0.234** | 0.775 | 2.782 | 5.534 | 0.464 |
| Watercraft | 0.258 | 0.452 | 0.179 | 0.429 | **0.153** | 0.248 | 0.900 | 1.871 | 3.405 | **0.232** |

Besides the Chamfer distance and normal consistency used in mesh evaluation (see Section 5.3 of our paper), we also list the 3D IoU scores [8] in Table 3.

Table 2: Quantitative comparisons on mesh reconstruction.

| Category | Chamfer Distance-$L_2$ ($\times 1000$) $\downarrow$ | | | | | Normal Consistency $\uparrow$ | | | | |
| | DMC | ONet | IF-Net | P2P-Net* | Ours | DMC | ONet | IF-Net | P2P-Net* | Ours |
|---|---|---|---|---|---|---|---|---|---|---|
| Bench | 0.312 | 0.857 | 0.428 | 0.180 | **0.125** | 0.772 | 0.743 | 0.791 | 0.780 | **0.839** |
| Watercraft | 0.363 | 1.152 | 0.619 | 0.148 | **0.092** | 0.794 | 0.766 | 0.815 | 0.835 | **0.898** |

Table 3: Average 3D IoU on object categories (%)$\uparrow$

| Category | DMC | ONet | IF-Net | P2P-Net* | Ours |
|---|---|---|---|---|---|
| Airplane | 33.45 | 55.25 | **72.95** | 68.15 | 69.07 |
| Rifle | 29.33 | 51.37 | 30.77 | 60.81 | **66.38** |
| Chair | 24.75 | 39.45 | 59.90 | 57.12 | **66.31** |
| Lamp | 22.28 | 43.59 | 60.25 | 56.15 | **73.27** |
| Table | 24.35 | 35.93 | **69.80** | 56.38 | 57.92 |
| Bench | 26.89 | 51.65 | **68.27** | 45.22 | 48.85 |
| Watercraft | 26.78 | 48.90 | 73.34 | 62.45 | **74.52** |
| Average | 26.83 | 46.59 | 62.18 | 58.04 | **65.19** |

## 4.2 Discussions on Normal Estimation

In this part, we mainly discuss the effects of using skeletal points impacted on point normal estimation. To this end, we design two configurations of networks. Since P2P-Net [14] shows promising results on both point and mesh completion, we adopt P2P-Net as the baseline method. As the original P2P-Net does not take account the normal estimation, we augment P2P-Net with extra channels for normal regression (see Section 5.3 in our paper). The second configuration is our SK-PCN without surface adjustment (Ours*), which is to investigate the effects of using shape skeletons. All networks produces 8,192 points for each object (inline with Section 5.3 in our paper). We present the CD and Normal Consistency scores on the chair category in Table 4, and the visualizations in Figure 8. Figure 8b shows the PSR results using normals directly calculated from our point clouds with [12]. The results in Table 4 indicate that skeletal points benefit both the point approximation and normal estimation.

Table 4: Ablative comparisons on mesh reconstruction.

| | Chamfer Distance-$L_2$ ($\times 1000$) $\downarrow$ | | Normal Consistency $\uparrow$ | |
| | P2P-Net* | Ours* | P2P-Net* | Ours* |
|---|---|---|---|---|
| Score | 0.258 | **0.177** | 0.801 | **0.847** |

| (a) Input | (b) PSR | (c) P2P-Net* | (d) Ours* | (e) GT |

Figure 8: Reconstruction results with different configurations.

## 4.3 Discussions on the Effects of Different Losses.

In this section, we mainly discuss the effects of the normal loss, adversarial loss and repulsive loss to the results of point completion. We use the P2P-Net as the baseline method and output

2048 points evaluated with CD and EMD metrics on 'chair' category to make the comparison. We augment P2PNet with extra dimension (see Section 5.3 in the paper) to estimate point normals (i.e. P2P-Net+normal loss), and extend it with our adversarial module to (P2PNet+normal&adversarial loss). Repulsive loss is added to all the methods. The (CD×e4↓, EMD×e2↓) values of the original P2P-Net are (2.94, 3.13), and the others achieve (2.98, 3.19) and (2.76, 1.70) respectively, while ours are (2.55, 0.49). The results indicate that normal loss is for point normal estimation, but unlike the adversarial loss, it can not help the point estimation.