[Reviews · NeurIPS 2020]

Review 1

Summary and Contributions: This paper proposed a point cloud completion network, which significantly outperforms existing encoder-decoder-based methods. Instead of generating the completed point cloud or the missing part of the input point cloud from a latent feature extracted, they predict the complete 3D skeleton points of input first, and then use the feature learned from input point cloud together with predicted 3D skeleton points to generate displacements from the skeleton points to surface points to complete the input point cloud. To get a better result, an attention module is used to bridge the local features of points into skeleton points. And a surface adjustment module is introduced to do a surface adjustment to make generated surface points similar with local input surface points.

Strengths: The result shown in this paper makes good progress in point cloud completion, and the idea of generating skeleton points for further point cloud completion is novel.

Weaknesses: It is not clear that whether the good results come from the surface points generated from skeleton points or the good design of the whole completion pipeline.

Correctness: Yes

Clarity: Yes

Relation to Prior Work: Yes

Reproducibility: No

Additional Feedback: All in all, the authors have done a good job. But the whole system is complicated and I think it is nontrivial to reproduce this work. It will be helpful if the source code is released (with dataset and other utils code). Previous works mainly complete point cloud by generating a coarse output from a latent feature and refine it, or deform a template point cloud to form the desired (missing) shape, while this work tries to generate skeleton points and then learn a displacement from the skeleton points to generate surface points and obtains good improvement. It is normal to think that the skeleton points generating module will play a most important role in this paper (Skeleton-bridged Point Completion). But the problem is, if the skeleton-bridged manner itself can get a good result. It is not clear that if the result of “Table 3: Comparisons between ablated versions” and “Table 2: Quantitative comparisons on mesh reconstruction” are obtained in the same setting (I notice that these two numbers (CD= 0.175 and EMD= 0.499) are the same in the two tables). If so, the skeleton-bridged manner itself cannot beat MSN, PF-Net, etc. (w.o. Non-Local Attention & w.o. Surface Adjustment, it will be CD=0.340 and EMD=2.261), while it is the attention module and surface adjustment can improve the result significantly. The necessity of these two modules is obvious, but the necessity of why using skeleton points generating module is unclear. It is very necessary to give such an evaluation: replace the skeleton points generated with a coarse point cloud and see if or not we can still get similar results. Another concern is, how good the skeleton points generating module is? How well will these points approximate the skeleton points? The result in the middle of Fig.1 looks a bit messy and uneven. Please show the skeleton points result and the ground-true result on more examples and provide quantitative analysis. Post-rebuttal: Thanks the authors for your careful rebuttal. It indeed addressed some of my main concerns, so I lean to accept. 


Review 2

Summary and Contributions: This paper proposes a novel method for 3D point cloud shape completion. The central ideas are 1) to reconstruct a full structural skeleton of the shape first; 2) to combine the global feature from the skeleton and the local feature from the partial point cloud input for faithful completion; 3) to leverage a patch GAN to refine the point cloud surface reconstruction. The experiments are very solid in proving the effectiveness of the proposed method compared to previous methods.

Strengths: 1) The central idea of predicting full shape structural skeleton first is novel to the shape completion task. Experiments show this structural backbone helps to provide global guidance for the completion and non-local details transferring. 2) The combination of global features from the reconstructed skeleton and the local features from the partial input point clouds is essential to generate more faithful reconstruction that preserves the visible parts. 3) Experiments are very solid, comparing to five previous state-of-the-art methods and beat all of them by good margins. Also, the qualitative figures show convincing results that match the numbers.

Weaknesses: 1) There are many modules in the proposed method. So, it's unclear if the baseline methods will improve a lot to be comparable to the proposed method if you use the same losses (e.g. the normal loss, the repulsive loss and the adversarial loss). Also, does using normal loss mean that your method has more supervision than the baseline methods? 2) Do you see failure cases that the skeleton fails to recover faithful structure to the input partial scan? Such a multi-stage pipeline may perform worse if the first-step structure prediction is of low quality. Is this true? It would be better to explain more failure cases.

Correctness: The claims and method are correct and the experiments are very solid.

Clarity: The paper is well written and easy to follow.

Relation to Prior Work: The discussion on previous and related works are adequate.

Reproducibility: Yes

Additional Feedback: Post-rebuttal: I remain the recommendation to accept this paper.


Review 3

Summary and Contributions: This paper introduces a skeleton-bridged point completion network for shape completion. The key idea is to first predict 3D skeleton from an input partial scan and then complete the surface by learning displacements from skeletal points. The main motivation of the paper is to decouple the shape completion into structure estimation and surface reconstruction so as to ease the learning difficulty and obtain surface details.

Strengths: - The overall pipeline with skeleton prediction + surface regression makes sense in general for point cloud completion. - The proposed method outperforms existing point cloud completion and mesh completion methods on ShapeNet benchmark. - Ablation study is provided.

Weaknesses: - The overall novelty is incremental. The main contribution mentioned in L62, i.e. recover skeleton + shape refinement, has been studied in previous works as mentioned in the related work. In my opinion, the fact that this work learns skeleton from partial scan, compared to learning skeleton from other modality [31], does not form a strong contribution. - It would be much better if a figure showing the whole pipeline is given. Fig 3, 4 are not easy to follow. -The ablation study is very helpful in understanding the effectiveness of the proposed components but not sufficient. I would suggest to provide the performance of the model with each component removed. Also, it is better to provide justification for the detailed design of modules in Fig 3 and 4 if they are not extended from previous work. - It is better to provide experiments on real dataset in order to understand the robustness against real data noise.

Correctness: Yes

Clarity: Yes

Relation to Prior Work: Yes

Reproducibility: Yes

Additional Feedback:


Review 4

Summary and Contributions: This paper presents a point completion network, namely SK-PCN, which is guided and bridged by meso-skeleton. The model is composed of skeleton generation and surface point completion parts and trained in an adversarial manner. In terms of contribution, the paper provides a new perspective of point completion guided by skeleton which is not seen in previous papers. The method has the potential to improve our understanding of neural network by replacing latent feature maps with an explicit skeleton representation. Furthermore, it shows SOTA performance on ShapeNet dataset.

Strengths: The paper is written clearly and well motivated. I like the idea of decoupling point completion into skeleton estimation and surface completion. An explicit intermediate representation has the potential to increase our understanding of neural network. The paper gives clear motivation on using seletion to bridge two parts of the task, and the introduction of Non-Local Attention. The theoretical claims are also reasonable. Extensive experiments and ablation studies are conducted to validate the overall performance and to test the newly introduced components of the model.

Weaknesses: Review Update: The authors' respond addressed my concern about using adversarial loss and skeleton performance. I would like to see those changed addressed in the final revised version. I'm happy with the ideas in the paper and I would like to see it published. ---------------------------------------------------------- 1. Although paper claims the importance of preserve the geometry on the observable region, I do not see clear motivation in using the adversarial training to achieve this. It seems that refining the output scan with observable input scan and training with CD loss can also achieve this objective. Using adversarial training can introduce extra learning burden. 2. Used as a bridge, the quality of skeleton can highly determine the final performance of surface completion. I would really like to see some qualitative demonstration of generated skeleton and more discussion on how the quality of skeletion can affect the final generation. With no direct supervision, can you guarentee the model is really learning a skeleton or something else?

Correctness: I don't see obvious errors in the claims or assumptions.

Clarity: The paper is well-written.

Relation to Prior Work: Yes.

Reproducibility: Yes

Additional Feedback:

[Author Response · NeurIPS 2020]

# Skeleton-bridged Point Completion: From Global Inference to Local Adjustment

We thank all reviewers for the time and expertise they have invested in the comments. We really appreciate their
recognition of our work on the method novelty, experiment performance and writing clarity. We sort out our responses
(**R**) to the comments (**C**) as follows. Hope they can address reviewers' concerns.

**C1.1:** *1) "Only using skeleton module (baseline of the ablation study) cannot beat MSN, etc." 2)*
*"It is necessary to replace the skeleton generated with a coarse point cloud to see the difference."*
**R1.1:** Thanks for this advice. 1) It is because this baseline completes surface points only using
predicted skeletons (see Sec 5.4). Details from input scans could be lost. We devise this baseline
to instigate how much the other modules leverage the input to improve the results. 2) We made
this ablation on 'chair' category (see Figure 1). The (CD↓, Normal Cons.↑) values are (2.96e-4,
0.81) and our (1.59e-4, 0.86). We think the reason could be that: coarse point cloud is still a
type of surface points. While skeletal points keep compact topology of the shape without surface
details. Using it as a bridge makes our method easier to recover complex structures. We will discuss it in detail.

Figure 1: Comparing with coarse points (right: ours).

**C1.2:** *"Please show the skeleton points result and the ground-truth and provide quantitative analysis."*
**R1.2:** Thanks for the suggestion. We output some skeleton results (2,048 points) in Figure 2 and will put them in the
final version. The average CD and EMD values to the GT are 2.98e-4 and 1.44e-2. Our codes will also be released.

**C2.1:** *"Whether the baseline methods will improve to be comparable to this method if using the*
*same losses? Also, does using normal loss means more supervision than the baseline methods?"*
**R2.1:** We keep their original loss because some methods (ONet, DMC) adopt implicit functions
to represent shapes, which do not support point losses, and some methods use similar losses with
us (PF-Net). In Sec 5.3 and supp. material, we have augmented the baseline P2PNet with our
modules + repulsive and normal losses (P2PNet*) to see the difference with us. The normal loss
indeed means an extra supervision but to supervise normal estimation. We augment P2PNet with
our modules to (P2PNet+normal loss) and (P2PNet+normal&adversarial loss) on 'chair' category.
Repulsive loss is added for each. The (CD×e4↓, EMD×e2↓) values are (2.94, 3.13), (2.98, 3.19),
(2.76, 1.70) respectively, and ours are (2.55, 0.49). We cannot see improvements in CD/EMD involving the normal loss.

Figure 2: Skeleton results. l->r: input, prediction, GT

**C2.2:** *"Such a multi-stage pipeline may perform worse if the first-step prediction is of low quality. Is this true?..."*
**R2.2:** Indeed the skeleton quality would affect surface results, but since our network is trained jointly, subsequent
modules can optimize the skeleton deviations and produce optimal results. We will put failure cases in the final version.

**C3.1:** *"In my opinion, the fact that this work learns skeleton from partial scan, compared to learning skeleton from*
*other modality [31], does not form a strong contribution."*
**R3.1:** The meso-skeleton provides a topology-consistent shape abstraction that inspires us to learn surface completion
bridged by skeletal points. As mentioned in other reviews, it is a novel attempt and achieves SOTA results. [31] also
adopts skeletal points but for another task (single-view reconstruction). Their image input presents totally different
modality with our sparse and irregular partial points. It thus requires us to tailor a unique method for partial-to-full
point completion, which is inherently different from their image-skeleton-voxel reconstruction with an encoder-decoder
network. Indeed we share an intuitive insight that skeleton can provide a global structure. For shape completion,
however, the major bottleneck lies on inferring the complicated topology from irregular points, where the meso-skeleton
presents a distinct advantage. The experiments also verified the worthiness of this first attempt. As mentioned by
reviewer#4, we also believe this work can provide a new perspective of point completion guided by shape skeleton.
**C3.2:** *"It would be much better if a figure showing the whole pipeline is given. Fig 3, 4 are not easy to follow."*
**R3.2:** Thanks for this comment. The whole pipeline is illustrated in Figure 2 (in our paper), and the detailed layer
information and data flow are demonstrated in the supplementary file. We will mention this in the paper to make it easier
to follow. Figure 3 and 4 are explained in Section 3.2 and 3.3, which will be further detailed in the revised version.
**C3.3:** *"I would suggest to provide the performance of the model with each component removed..."*
**R3.3:** Sec 5.4 presents the ablation study on our main modules. We will further ablate the skeleton module as in R1.1.
**C3.4:** *"It is better to provide experiments on real dataset in order to understand the robustness against real data noise."*
**R3.4:** Thanks for the advice. We test our network on real scans in Figure 3. More results will be put in the final version.

**C4.1:** *"Although paper claims the importance of preserve the geometry on the observ-*
*able region, I do not see clear motivation using the adversarial training..."*
**R4.1:** Before considering the adversary loss, we actually refined the scan with observ-
able points only using CD loss. However, we observed uneven point distribution on
the observable area as discussed in PU-GAN. Thus we adopted the adversarial loss in
PU-GAN to improve the visual quality on surfaces (see Section 3.3).
**C4.2:** *"More qualitative results of generated skeleton and discussions on its effects..."*
**R4.2:** Thanks for this suggestion. Here we list some results in Figure 2. For the page

Figure 3: In-the-wild tests. l->r: input, points, mesh.

limit, we will analyze its effectiveness in the revised version. Our current skeleton generation is designed for supervised
learning. We hope it can lay a foundation for the future work in unsupervised skeleton learning.

[Meta-Review · NeurIPS 2020]

The reviewers felt that this is a strong submission with an interesting idea, extensive experiments and ablations, strong results, clear writing and clear motivation. Some reviewer concerns involve missing ablation to understand the contribution of each of the losses; missing analysis of the inferred skeleton; and some reduced novelty compared to [31]. These concerns were addressed in the rebuttal. We highly encourage the authors to update their manuscript to reflect these points.